# Experimental Study on PVA-MgO Composite Improvement of Sandy Soil

**DOI:** 10.3390/ma15165609

**Published:** 2022-08-16

**Authors:** Zhongyao Li, Zhewei Zhao, Haiping Shi, Jiahuan Li, Cheng Zhao, Peiqing Wang

**Affiliations:** 1Water Conservancy Civil Engineering Department, Tibet Agriculture and Animal Husbandry College, Linzhi 860100, China; 2Tibet Construction Water Conservancy and Electric Power Engineering Technology Research Center, Linzhi 860100, China; 3Key Laboratory of Geotechnical and Underground Engineering of Ministry of Education, Tongji University, Shanghai 200092, China

**Keywords:** active magnesium oxide, polyvinyl alcohol (PVA), mechanical strength, microstructure

## Abstract

Materials with violent hydration reaction such as cement are used to solidify sandy soil slopes, which will cause destructive damage to the ecology of the slopes. In this paper, polyvinyl alcohol (PVA) and activated magnesium oxide (MgO) are used to improve sandy soil, and the effects of the dosage and curing age of modifiers on the mechanical properties of solidified sandy soil are studied. The dry–wet durability of the composite improved sandy soil is analyzed using a dry–wet cycle test, and the improvement mechanism of PVA and activated magnesium oxide is revealed using an electron microscope. The results show that the curing effect of polyvinyl alcohol and activated magnesium oxide on sand particles is better than that of polyvinyl alcohol alone. The compressive strength of improved soil samples increases with the increase of curing time, and magnesium oxide as an improved material needs appropriate reaction conditions to give full play to its role. The compressive strength of composite improved samples increases first and then decreases during the dry–wet cycle. Through the observation of microstructure, it can be seen that the cementing material wraps and connects the sand particles, and the cementing material of the sample after the dry–wet cycle develops more completely; if the magnesium oxide content is high, cracks may appear inside the sample.

## 1. Introduction

The stability of sandy soil slope is poor under rainfall and other conditions [1]. Therefore, it is necessary to take engineering technologies to reinforce the slopes. Soil improvement technology has been widely used in road engineering, water conservancy engineering, slope treatment engineering and other fields. In engineering, the choice of improved materials is mostly limited to traditional improved materials such as cement, lime and fly ash, and their compound addition [2,3,4]. However, the traditional improved materials have many disadvantages in the field of soil improvement (such as the impact on ecological environment, etc.) With the development of new building materials, more and more polymer materials and biomass materials are used as soil improvers [5,6,7,8,9,10,11,12], which effectively improve the engineering performance of soil. Under suitable conditions, biomass materials can induce the deposition of carbonate and fill the defects such as cracks [13,14,15]. Polymer can realize the process of hardening–dissolving–re-hardening under certain conditions due to the influence of its long chain structure, and it can aggregate to the defects to fill them [16,17,18]. All these show the effect that traditional materials such as cement cannot achieve in the field of self-repair of engineering materials.

Due to abundant rainfall in southeast Tibet, frequent periodic precipitation will make the soil unstable. On the other hand, the use of traditional curing agent will affect the ecosystem due to water circulation and the destruction of surface plants, thus reducing the stability of soil eroded by water [19,20,21,22,23,24,25]. Zhu et al. [26] combined the addition of hydroxypropyl methylcellulose (HPMC) with MICP technology to improve the surface layer of the soil. Their test results revealed that the rainwater scour loss and wind erosion loss were reduced. As a highly active material, active magnesium oxide has a good effect in soil improvement and reinforcement. Therefore, more and more researchers choose to improve the engineering properties of soil by using the cementitious material, such as activated magnesia [27]. In this paper, PVA and activated magnesia are used as improvement materials to carry out improvement tests on sandy soil, and the application effect of the interaction between organic polymer and highly active inorganic cementitious materials on sandy soil improvement is explored.

## 2. Materials and Methods

### 2.1. Materials

The coarse-grained soil used in the test was taken from the loose deposit beside landslide along the Palong Zangbo River in southeast Tibet, at a depth of 1.0~1.5 m. Geotechnical tests were carried out according to the “Standards for Geotechnical Test Methods” [28]. The grain size distribution curve is shown in Figure 1. The maximum grain size is 2.5 mm, *p* ≤ 0.075 mm = 5.68% (where *p* ≤ 0.075 mm means the cumulative mass of soil particles with a grain size of less than or equal to 0.075 mm accounts for the percentage of the total mass). There are many intermediate grain sizes in this soil, and more than 50% of the particles are concentrated in 0.63–1.25 mm groups.

The modified material is high polymer polyvinyl alcohol (PVA) 20–99 (produced by Yunsheng Chemical Company in Shanxi, China) and active magnesium oxide, among which the physical properties of PVA are shown in Table 1. The PVA is feather-shaped particles (as shown in Figure 2a) and is insoluble in cold water and soluble in boiling water to form a stable solution. A PVA solution with an effective substance content of 5.23% (as shown in Figure 2b) is prepared (if the effective substance is not mentioned separately in the following text, the PVA addition amount refers to the addition of the prepared PVA solution).

Active magnesium oxide is white amorphous powder (as shown in Figure 2c), odorless, tasteless and nontoxic, with a specific gravity of 3.58, a melting point of 2852 °C and a boiling point of 3600 °C. The particle diameter distribution is in the range of 0.14–40.15 μm, with median particle diameter of 4.52 μm, area average particle diameter of 2.06μm, volume average particle diameter of 6.95 μm and specific surface area of 1076.36 m^2^/kg.

### 2.2. Experiment Methods

#### 2.2.1. Unconfined Compressive Strength Test

Through unconfined compressive strength test, the improvement effect was determined. The effects of PVA dosage and curing age on the mechanical properties of solidified soil were mainly studied. PVA solution (6, 8, 10 and 12%) was added in groups according to the test plan summarized in Table 2. Three samples were made for each group by adding amounts, and the results were averaged within the error range. Cylindrical samples with φ × h = 50 mm × 100 mm were made using a static soil compactor. After the sample was prepared, it was cured under simulated natural conditions (temperature 20 °C, humidity 40%). The unconfined compressive strength of the sample was tested using an electric press with a precision of 0.01 kn, and the loading speed was controlled to be 1 mm/min during the loading process.

#### 2.2.2. Permeability Coefficient Test

The permeability of sandy soil is one of the important indexes. The permeability coefficient of improved soil was measured using a TST-55 vertical permeability instrument. The effects of pristine PVA and PVA-MgO composite modification on permeability were studied.

#### 2.2.3. Capillary Absorption

After curing for 7 days, the composite improved sample (φ × h = 50 mm × 100 mm cylinder) is put into 3 mm deep water, and the quality of each sample is measured with a balance with an accuracy of 0.01 g until the sample is completely saturated. The weight of the sample is weighed every 2 h during water absorption.

#### 2.2.4. Wet–Dry Cycle

After curing the samples for 7 days, the dry–wet cycle test was carried out indoors to simulate natural conditions. Of the samples that had completed the specified number of wet and dry cycles, one sample from each ratio was reserved for the subsequent SEM scanning test, and the remaining samples were subjected to an unconfined compression test using a triaxial testing machine.

The wetting–drying cycle test is divided into two processes of water absorption to saturation and air drying [29]. The saturation process of the samples is as follows:(1)After placing the permeable stone at the bottom of the container, add water to the container until the water surface is as high as the permeable stone and place the sample on the permeable stone.(2)Gradually add water until the sample is completely submerged in water and place the permeable stone at the top of the sample.(3)Add a certain amount of water to the container every 2 h to keep the water level in the container constant.(4)Saturate the sample after 24 h and remove it.

The drying process is as follows:(1)The samples were placed in the drying oven after saturated with water, and the temperature of the drying oven was set to 40 °C.(2)The samples were weighed every 4 h during the drying process to ensure that the moisture content meets the test requirements.(3)When the moisture content no longer changes, stop drying.

## 3. Results

### 3.1. Influence of Modified Material Content on Strength

Figure 3a shows the stress–strain curves of air-dried curing for 3 days with different PVA content. It can be seen that with the increase of PVA content, the unconfined compressive strength (UCS) of improved sand soil increases, and the UCS of 6%PVA content is 147.70 kPa. With the addition of 12%PVA, the UCS of the improved sand is only 320.86 kPa, and the strength is not obviously improved. With the increase of strain, the PVA elastic mesh between soil particles breaks, resulting in the decrease or non-increase of strength and the fluctuation of the curve. Figure 3b shows the stress–strain curves of air-dried sand after 7 days of curing with different PVA content. With the increase of curing time and PVA, also a complete network structure is formed to provide strength. The instability of the stress curve is weakened.

Figure 3c,d is stress–strain curves after curing for 14d and 28d, respectively. Similar to curve curing for 7 days, the curve is relatively smooth with less instability, which indicates that PVA has formed a complete structure in the improved sand, which can significantly enhance the mechanical properties of sand.

Figure 4a shows the curve of the UCS with curing time under different PVA content. After 3 days of curing, the highest strength of PVA modified sand is 320.86 kPa. As can be seen from the figure, the strength increases rapidly within 3–7 days. The strength of the modified sand with PVA content of 12% can reach 1038.96 kpa. The strength increased slowly in 7–14 days, and after 14 days, the strength of the modified sand with 12%PVA content was 1115.36 kpa. After that, the unconfined compressive strength of each content tends to be stable, and the unconfined compressive strength of 28 days is basically the same as that of 14 days. After 7 days of curing, the PVA modified sand basically completed the dehydration and drying process, and the strength of 12%PVA reached more than 90% of the strength of 28 days.

Figure 4b shows the curve of the UCS of improved soil and the content of PVA at different curing ages. It can be seen from the figure that the curve of curing for 3 days is far from that of 7 days, 14 days and 28 days, and the trend of strength with the increase of PVA content is more even. The positions of the three curves of curing, 7 days, 14 days and 28 days, are close, and the growth trend is the same.

Figure 5a shows the stress–strain curve of the composite improvement of PVA and magnesium oxide after curing for 3 days. The stress–strain curve is smooth and continuous as a whole, compared with only adding PVA as modifier. There is no sharp increase in the curve, and the peak value is gentle. Figure 5b is the stress–strain curve of the specimen cured for 7 days in the same coordinate system. After 7 days of curing, the stress–strain curve has a more obvious peak value, and the strength is obviously improved. The strain value corresponding to the peak value decreases slightly, indicating that the improved sample may be more brittle. Figure 5c is the stress–strain curve of the specimen cured for 14 days in the same coordinate system. After this curing time, the stress–strain curves of all the samples with different dosage increase obviously. Except for the curve of 12%PVA and 12%MgO, there is a significant distance between the other three curves. After curing for 28 days, the stress–strain curve is shown in Figure 5d. Compared with the 14-day curve, the 28-day curve is still obviously elevated, and the curves are smooth and close in position.

Figure 6 shows the curve of UCS of composite modified samples with curing time. Under the condition of curing for 3 days. The UCS of composite improvement of 12%PVA and 4% MgO reaches 487.74 kPa, and that of composite improvement of 12%PVA and 16% MgO reaches 830.15 kPa, which is higher than that of single improvement of 12%PVA (320.86 kPa). The addition of magnesium oxide makes the improved sand have higher strength in the early stage.

The curve of 12%PVA and 4% MgO shows a straight line of strength growth in the whole curing period, while the curve of other additives gradually slows down with the curing time. After 28 days, the strength of 12%PVA and 4% magnesium oxide has exceeded that of 12%PVA and 12% magnesium oxide. Different from PVA modified sand whose strength has reached 90% of 28-day strength after curing for 7 days alone, the strength of modified sand after adding magnesium oxide keeps increasing in the curing cycle. After 28 days of curing, the maximum strength can reach 1909.86 kPa, and the strength can reach 1527.89 kPa only by adding 4% magnesium oxide. It is 34–71% stronger.

### 3.2. Permeability

The permeability coefficient of sandy soil used in the experiment is 8.61 × 10^−3^, and the change of permeability coefficient of PVA-improved soil samples is shown in Figure 7a. When the content of PVA-improved soil samples is 6%, 8%, 10%, and 12%, the permeability coefficients are 8.42 × 10^−3^ cm/s, 8.37 × 10^−3^ cm/s, 8.27 × 10^−3^ cm/s and 8.61 × 10^−3^ cm/s, respectively. With the increase of polyvinyl alcohol content, the permeability coefficient decreases slightly, but the overall change range is very small.

The maximum content of PVA solution is 12% in the test, but the effective substance content of PVA is only 5.34%, that is, about 95% of the solution is water. With the increase of curing time, PVA will gradually remove water to improve the strength of the improved sample, and the remaining effective substance only accounts for about 0.6% of the soil weight, because polyvinyl alcohol will not generate new substances in the soil, and the soil strength will be improved only by the linkage between its macromolecules. Such a small amount of admixture obviously has no obvious effect on pore filling, and the change of permeability coefficient conforms to the law. In the later stage, we should consider adding polyvinyl alcohol solution with a higher content of effective substances, so that it can improve the strength of improved soil more obviously, and at the same time, it can also reduce the permeability of soil.

Figure 7b shows the change curve of permeability of the composite modified sample of PVA and active magnesium oxid.e. When the content of PVA is fixed at 12%, the permeability coefficient is 8.20 × 10^−3^ cm/s. With the increase of the content of active magnesium oxide (4%, 8%, 12% and 16%), the permeability coefficient of the sample decreases gradually, reaching 7.44 × 10^−3^ cm/s, 4.53 × 10^−3^ cm/s and 4.02 × 10^−3^ cm/s, respectively. This is because magnesium oxide is added to fill the pores between soil particles. 

### 3.3. Capillary Absorption

Figure 8 shows the curve of capillary water absorption weight and time of PVA-MgO composite modified samples. The absorption process of capillary water can be divided into three stages: the early stage of rapid rise, the middle stage of slow rise and the later stage of relative stability. After the addition of active magnesium oxide, the rising speed of capillary water is related to the amount of magnesium oxide. It can be seen that under the same amount of PVA, with the increase of magnesium oxide, the water absorption of the modified soil sample first decreases and then increases. When the amount of active magnesium oxide is 8%, the water absorption rate is the fastest, and the total amount of water absorption is the lowest.

### 3.4. Wet–Dry Cycle

The improved composite sample is very dense after being soaked in water, and there is no surface cracking or peeling of the sample (as shown in Figure 9). Figure 10 shows the curve of the relationship between unconfined compressive strength(UCS) and the number of dry–wet cycles of composite modified samples with 12%PVA and different dosage of active magnesium oxide. The change trend of the curve between the unconfined compressive strength and the number of dry–wet cycles at various dosages is basically the same. With the number of dry–wet cycles increasing, the UCS of modified soil samples increases first and then decreases. After 1 to 5 dry–wet cycles, the compressive strength of modified soil samples generally increases, and the increase is all large.

This is because the active magnesium oxide added in the composite modified sample did not completely react after curing for 7 days. The soaking process and air-drying temperature of 40 °C in the dry–wet cycle also provided an ideal temperature for the reaction to promote the reaction. After many soaking and air-drying processes, the active magnesium oxide in the composite modified sample gradually changed into basic magnesium carbonate, and the compressive strength of the sample increased first in this process. After reaching the peak value, the internal structure of the sample was destroyed due to the further reaction of excess active magnesium oxide, and the compressive strength of the sample decreased due to the combined action of water erosion in the dry–wet cycle, until it basically reached stability after 20 dry–wet cycles. 

It can be seen from Figure 10 that the modified sample of 12%PVA + 4% active magnesium oxide reaches the peak value of 2327 kPa after 5 times of wet and dry cycles, which is 1609 kPa higher than that of 7 days of curing, and its strength is 324.09% of that of 7 days of curing. The modified sample of 12%PVA + 8% active magnesium oxide only reaches the peak value of 3219 kPa after 10 times of dry–wet cycles, which increases by 2083 kPa compared with that after 7 days of curing. The strength is 283.36% of that after 7 days of curing, and the strength increases slowly. The modified sample of 12%PVA + 12% active magnesium oxide reached the peak value of 2445 kPa after 5 times of wet–dry cycles, which increased by 1022 kPa compared with that after 7 days of curing, and its strength was 171.82% of that after 7 days of curing. The strength of the sample modified by 12%PVA + 16% active magnesium oxide reaches the peak value of 3652 kPa after three times of dry–wet cycles, which is 2384 kPa higher than that of the sample cured for 7 days, and the strength is 288.01% of that of the sample cured for 7 days. The peak time and the increase range are different from the dry–wet cycles under different compound modified dosages.

After 20 wet and dry cycles, the strength decreased by 20.57% of the peak strength under the compound improvement of 12%PVA + 4% active magnesium oxide. The strength of the sample modified by 12%PVA + 8% magnesium oxide increased and decreased slowly, and the peak value was much higher than that of the composite modified by 12%PVA + 4% activated magnesium oxide. After 20 wet–dry cycles, the strength decreased by 15.03% of the peak strength, and the sample was weakly affected by the strength decrease caused by the continuous reaction of activated magnesium oxide. The strength of 12%PVA + 12% MgO improved sample increased rapidly in the early stage, but the strength decreased obviously, after 5 times of dry–wet cycles, only exceeding that of 12%PVA + 4% MgO improved sample at the peak, and then, the strength decreased to the lowest among the four admixtures. After 20 times of dry–wet cycles, the strength decreased by 28.54% of the peak strength. After the compressive strength of 12%PVA + 16% MgO improved sample reached the peak, after 20 dry–wet cycles, the strength became 2602.5 kPa, with a decrease of 28.73%. The large decrease of strength was mainly due to the further reaction of the excessive doped active magnesium oxide, which led to the destruction of the structure of PVA and the finished basic magnesium carbonate in the sample, resulting in the decrease of strength.

### 3.5. Microscopic Analysis

Figure 11, Figure 12 and Figure 13 are scanning electron microscope images of composite modified samples of polyvinyl alcohol and activated magnesium oxide, among which, Figure 11 and Figure 12 are samples after drying and wetting cycles, and Figure 13 is samples cured at the same age without drying and wetting cycles. It can be seen from the figures that the boundary of soil particles is blurred, and the pores among soil particles are filled by the modified materials added in the test range.

Figure 11a is the scanning electron microscope image of the composite modified sample with 12%PVA + 4% active magnesium oxide after 500 times magnification. It can be seen from the figure that the modified material has incomplete filling of pores, and there are pores and weak points in the cementing material and between the cementing material and sand particles, so the cementing effect on the modified sample is limited.

Figure 11b is a 2000-fold magnified picture of the composite modified sample with 12%PVA + 4% active magnesium oxide. From the picture, it can be seen that the three-dimensional structure generated by the composite cementing material of active magnesium oxide and PVA exists only in a small amount, but the three-dimensional structure exists in the cementing material. However, most of the cement is not fully developed, and there are obvious pores and weak points in the cementing material.

Figure 12a shows the SEM image of the sample with 12%PVA + 8% active magnesium oxide after 500 times magnification. It can be seen from the figure that although there are still some pores among the soil particles, there are dense and uniform cementing substances among the soil particles, which connect the sand particles forming a whole. Sand particles are wrapped by cementing material, and the boundary disappears.

Figure 12b shows the sample with 12%PVA + 8% active magnesium oxide content. After 2000 times magnification, it can be seen at high magnification that the cementing material generated by PVA and active magnesium oxide is fully developed and presents a uniform grid shape. Compared with 4% active magnesium oxide content, the amount of basic magnesium carbonate crystals as the reaction product is more and uniform. When the stress of the sample is improved, the uniform cementing material can effectively share the stress of the sample.

Figure 13a shows the scanning electron microscope image of the sample after curing for 7 days with 12%PVA + 8% active magnesium oxide, which has not been subjected to wet–dry cycle and has been magnified 500 times. Although it can be seen in the figure that there are cementing materials among sand particles, which wrap the sand particles, and there is no obvious boundary between them, no compact cementing materials are formed in the composite improved sample after curing for 7 days under natural conditions, and the mixture of PVA and active magnesium oxide is loose in the sample.

Figure 13b shows the scanning electron microscope image of the sample with 12%PVA + 8% active magnesium oxide after curing for 7 days, which has not been subjected to wet and dry cycles. Under this magnification, it can be clearly seen that PVA and active magnesium oxide exist among sand particles, and there are a lot of pores and defects in the cementing material formed by them.

Figure 14a shows the sample with 12%PVA + 12% active magnesium oxide. After 150-fold magnification, the sand particles in the sample are tightly wrapped by cementing material, and the sand particles can’t be distinguished in the picture. However, it can be clearly seen from the figure that cracks exist. The cracks are caused by the volume expansion caused by the reaction of active magnesium oxide between the sand particles, which damages the cementing materials that have been completely reacted in the early stage. Figure 14b is the scanning electron microscope image of the sample with 12%PVA + 12% active magnesium oxide content, which is magnified by 500 times. It can be clearly seen from the image that the shapes of the two ends of the crack are consistent, and it can be determined that the cementing material reacts first to form the whole structure, and then the internal active magnesium oxide continues to react to destroy the structure formed in the early stage, resulting in visible cracks, resulting in the reduction of the overall strength of the sample. The composite improvement strength of the sample with this content is not as good as that with less content of 12%PVA + 8% active magnesium oxide.

Figure 15 is the scanning electron microscope image of the sample with 12%PVA + 16% active magnesium oxide after 500 times magnification, respectively. Under this dosage, the visible range is filled with cementing material, which tightly wraps the sand particles, and there are no sand particles in the field of vision. The resulting cementing material is very dense, and there are still a few cracks in the cementing material, but the width and length of the cracks are obviously less than the improvement under the dosage of 12% active magnesium oxide.

## 4. Discussion

From the perspective of improvement effect, both PVA and the composite action of PVA and magnesium oxide can effectively improve the strength of sandy soil. The addition of PVA as the main improved material is also small. In some studies, the addition of PVA as the improved material is about 1% [30]. In this paper, the addition of PVA effective substance is about 0.6% of the sand weight. In some studies of cement-improved soil [31,32,33,34], the strength of the improved soil with 10% cement content after 28 days maintenance is about 1200 kPa or 500 kPa. Although the price of cement is cheaper than that of PVA, in a certain range, the strength of cement-improved soil can be exceeded when 1/10PVA is used to improve soil. This is economical.

In the process of curing, PVA mainly loses water, so that it can cling to the surface of sand particles. Due to the influence of sample volume and shape, the internal water loss rate is different, as shown in Figure 16. A more suitable curing method should be selected to make the moisture content of the improved sand uniform, instead of just waiting for a longer time. If the dehydration process can be completed in a shorter time, the strength will meet the requirements in a short time.

## 5. Conclusions

The unconfined compressive strength increases with the increase of PVA additions. The strength of 12%PVA is the highest, which can increase from 25.46 kPa to 1115.36 kPa.With the increase of curing time, the unconfined compressive strength of polyvinyl alcohol, polyvinyl alcohol and active magnesium oxide composite modified samples all increased, and the strength of polyvinyl alcohol composite modified samples remained basically stable after curing for 14 days, but the composite modified samples could not exert all the improved effects under air-drying conditions.The addition of active magnesium oxide increases the durability of dry–wet cycle. In the process of dry–wet cycle, the strength first increases and then decreases, which is due to the full reaction of active magnesium oxide in the process of dry–wet cycle. The continuous volume expansion of excess active magnesium oxide in the later period and the erosion of the dry–wet cycle reduce the strength. Therefore, the additional amount of active magnesium oxide should not be too large.As the effective substance in the PVA solution only accounts for 5.23%, which is about 0.6% of the sand weight, it has little effect on improving the permeability of sand. However, the activated magnesium oxide will slightly reduce the permeability of sand due to its volume expansion filling the pores during the reaction.It has good compatibility between PVA and magnesium oxide and can form a dense cementing material. Excessive magnesium oxide will cause cracks in cement due to continuous reaction.

## Figures and Tables

**Figure 1 materials-15-05609-f001:**
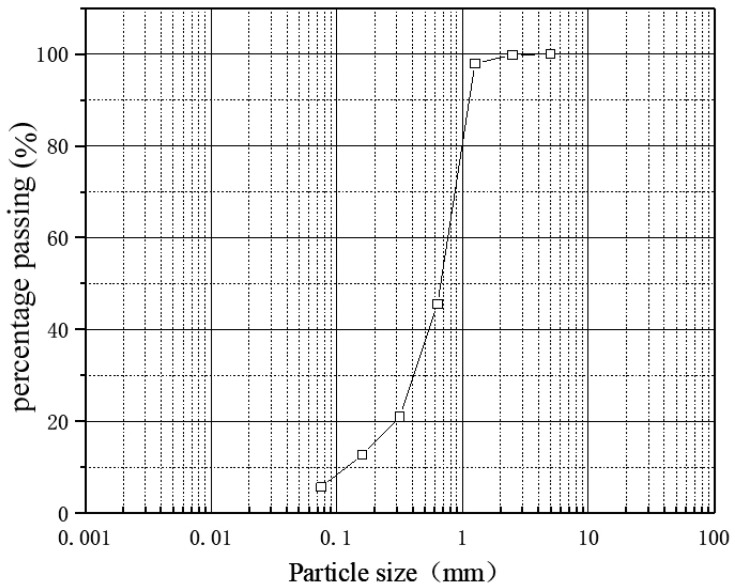
Grain grading curve of soil.

**Figure 2 materials-15-05609-f002:**
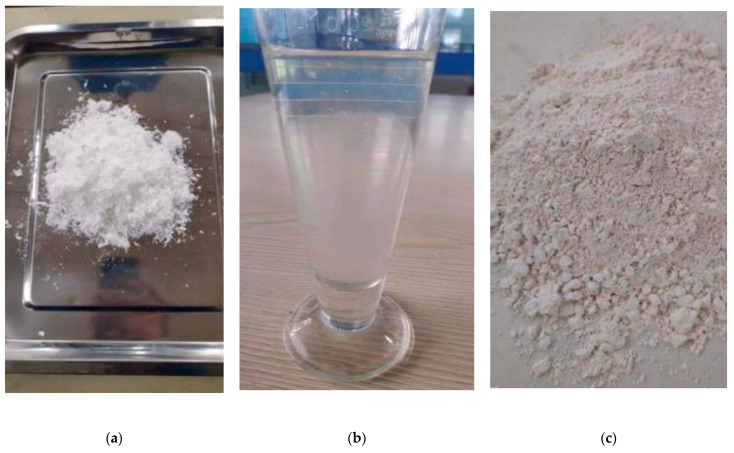
Improved materials: (**a**) PVA fiber, (**b**) PVA solution, and (**c**) active magnesium oxide.

**Figure 3 materials-15-05609-f003:**
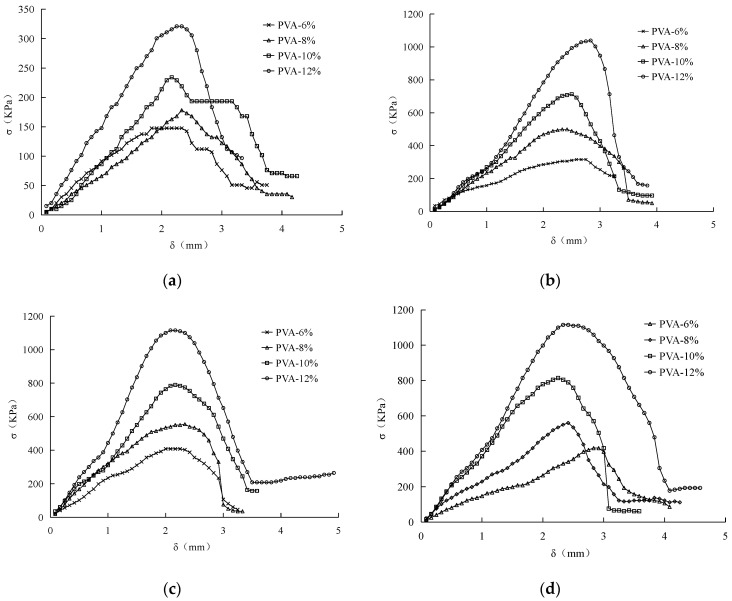
(**a**) 3 day; (**b**) 7 day; (**c**) 14 day; (**d**) 14 day. Stress–strain curves of PVA-improved soil under different curing time.

**Figure 4 materials-15-05609-f004:**
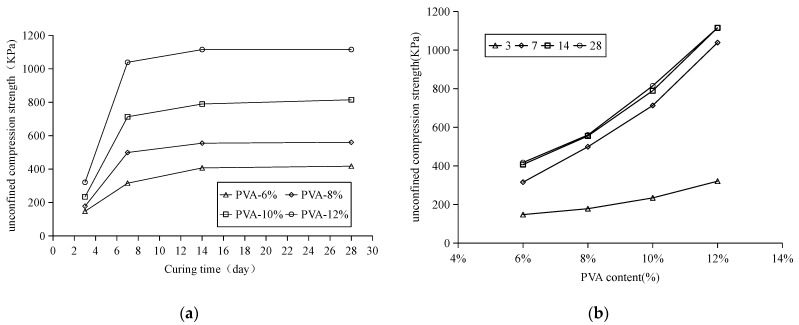
(**a**) UCS change curve with different curing time; (**b**) UCS change curve with different PVA content; Relationship between UCS and curing age of polyvinyl alcohol modified samples.

**Figure 5 materials-15-05609-f005:**
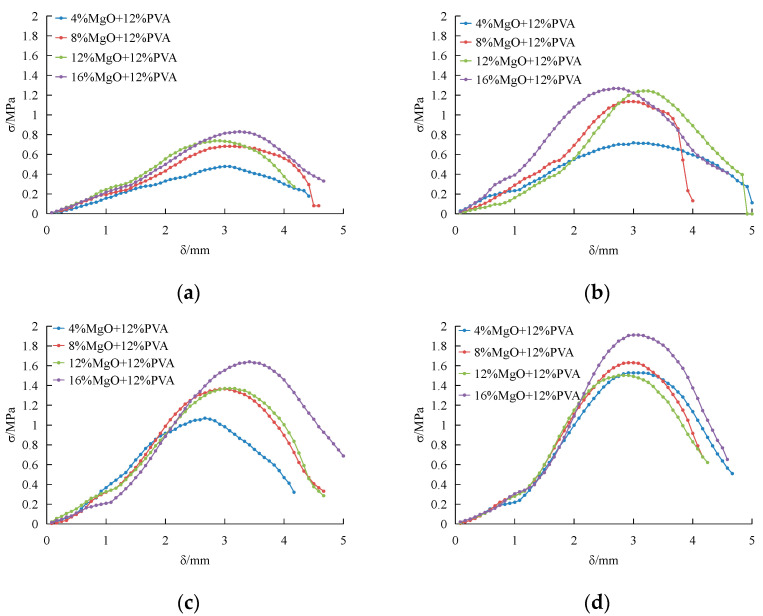
(**a**) 3 day; (**b**) 7 day; (**c**) 14 day; (**d**) 28 day. Stress–strain curve of PVA and activated magnesium oxide composite improvement.

**Figure 6 materials-15-05609-f006:**
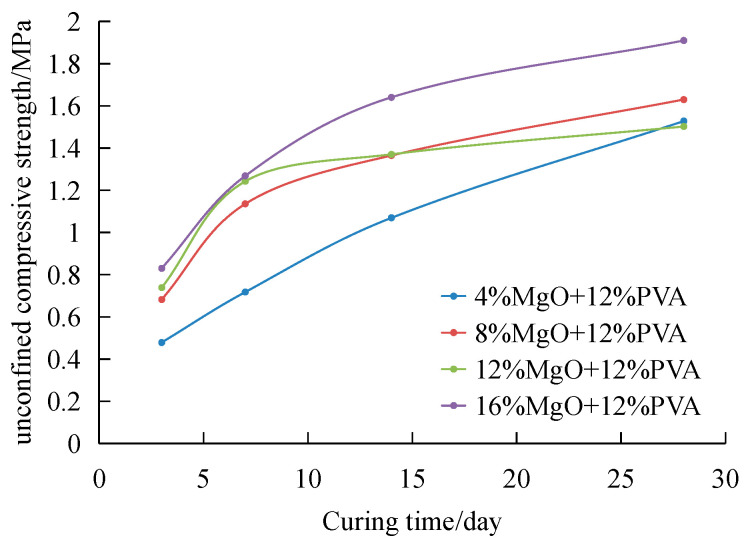
The change curve for UCS of composite modified samples with curing time.

**Figure 7 materials-15-05609-f007:**
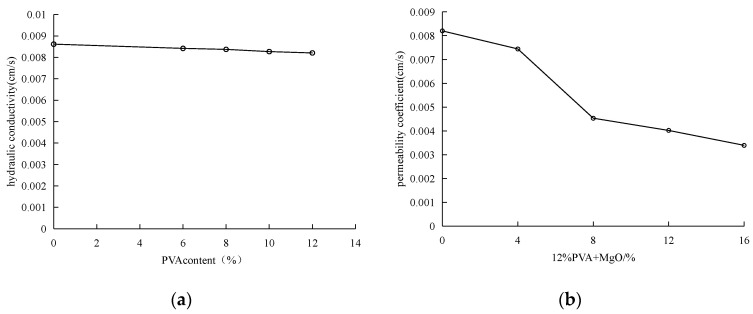
(**a**) Permeability coefficient curve of different PVA content; (**b**) Permeability coefficient curve of different MgO content; Variation curve of permeability coefficient.

**Figure 8 materials-15-05609-f008:**
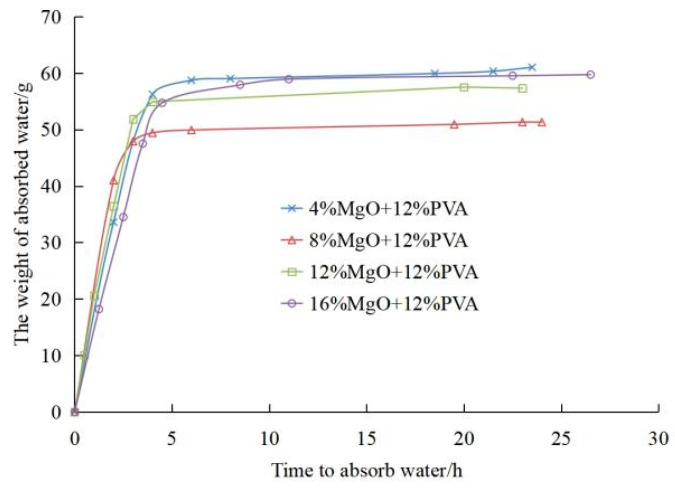
Absorption of capillary water.

**Figure 9 materials-15-05609-f009:**
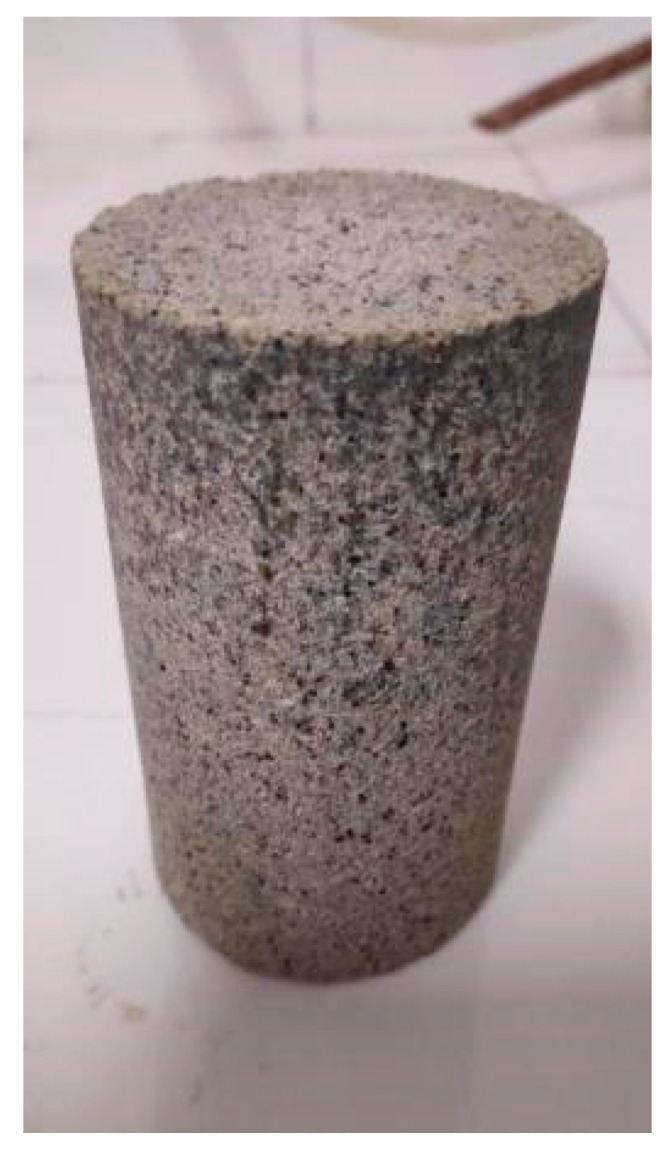
A sample that has undergone a wet–dry cycle.

**Figure 10 materials-15-05609-f010:**
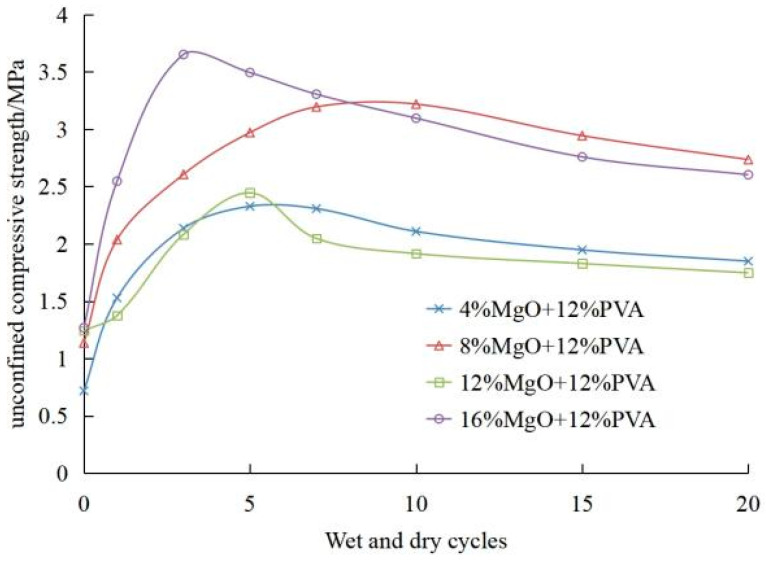
Curve of strength of sample with dry–wet cycle.

**Figure 11 materials-15-05609-f011:**
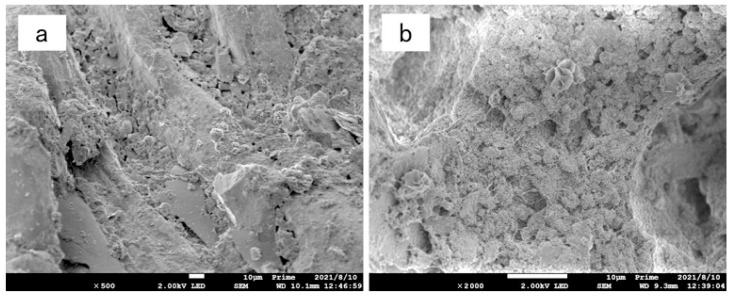
Electron microscopic picture of 12%PVA + 4% active magnesium oxide composite improvement. (**a**) 500-fold magnified (**b**) 2000-fold magnified.

**Figure 12 materials-15-05609-f012:**
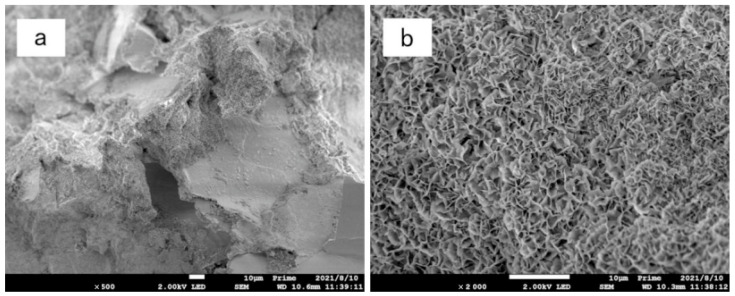
Electron microscopic picture of 12%PVA + 8% active magnesium oxide composite improvement. (**a**) 500-fold magnified (**b**) 2000-fold magnified.

**Figure 13 materials-15-05609-f013:**
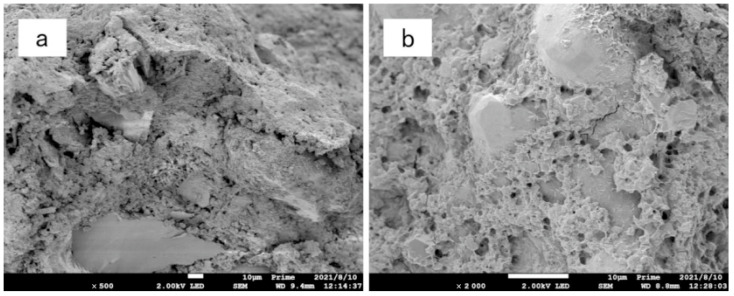
Electron microscopic image of the sample modified by 12%PVA + 8% active magnesium oxide without dry–wet cycle. (**a**) 500-fold magnified (**b**) 2000-fold magnified.

**Figure 14 materials-15-05609-f014:**
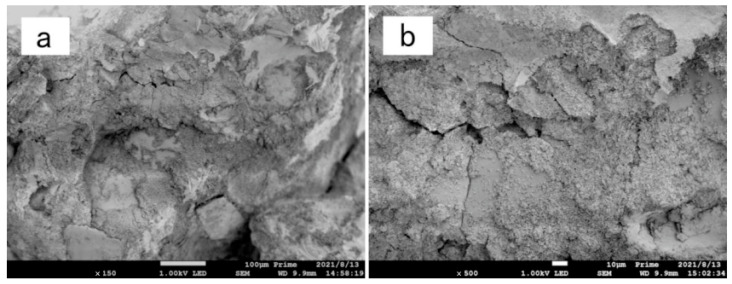
Electron microscope image of sample with 12%PVA + 12% active magnesium oxide content. (**a**) 150-fold magnified (**b**) 500-fold magnified.

**Figure 15 materials-15-05609-f015:**
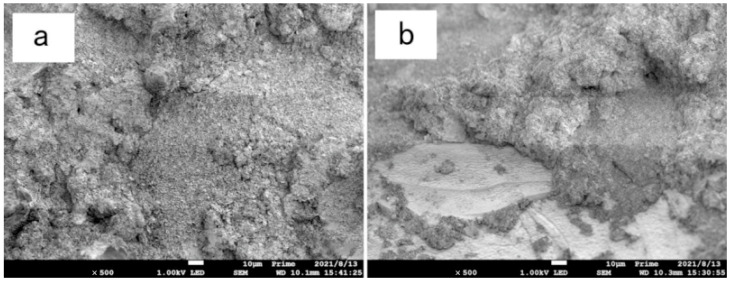
Electron microscope image of sample with 12%PVA + 16% active magnesium oxide content. (**a**) 500-fold magnified (**b**) 500-fold magnified(cement edge).

**Figure 16 materials-15-05609-f016:**
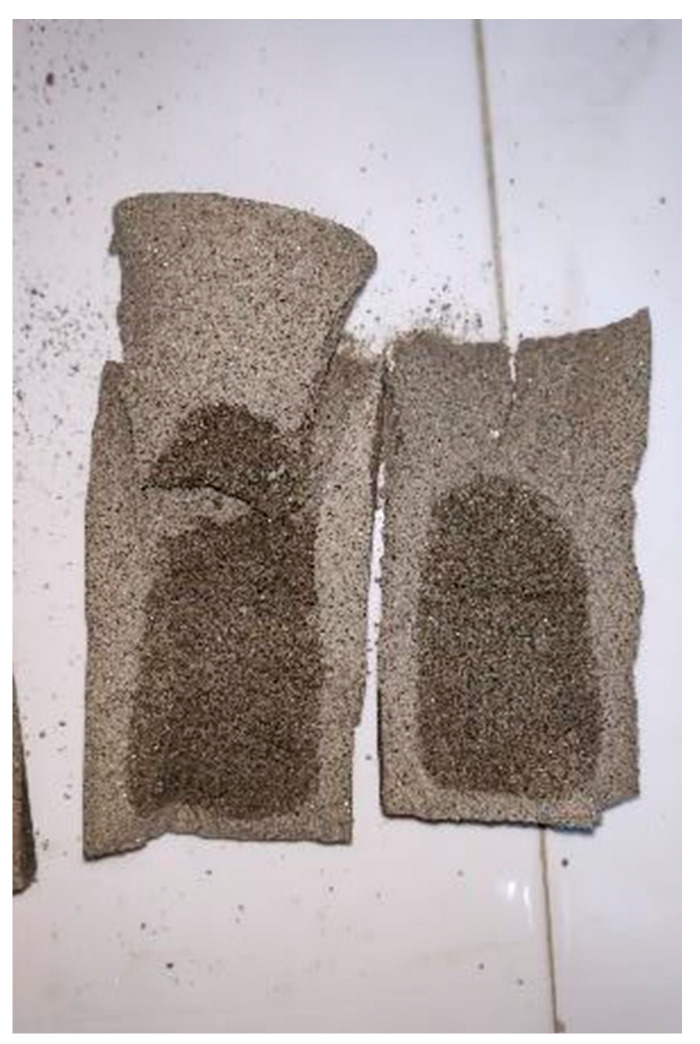
The area with uneven moisture content in the sample.

**Table 1 materials-15-05609-t001:** Basic properties of polyvinyl alcohol (PVA).

Category	Viscosity (mPas)	Alcoholysis Degree (%)	Volatile Matter (%)	Ash (%)	pH
20–99(H)	34.0–42.0	98.5–10 (%)	5	0.5	5–7

**Table 2 materials-15-05609-t002:** Details of test content.

Group	Number	PVA Content (%)	Magnesium Oxide Content (%)	Curing Time (d)	Number of Wetting-Drying Cycles
**Non-stabilized soil**	1	0	0	3, 7, 14, 28	0
**PVA alone**	3	6, 8, 10, 12	0	3, 7, 14, 28	0
	7	6	0	7	1, 3, 5, 7, 10
	8	8	0	7	1, 3, 5, 7, 10
	9	10	0	7	1, 3, 5, 7, 10
	10	12	0	7	1, 3, 5, 7, 10
**PVA and MgO**	11	12	4	3, 7, 14, 28	0
	12	12	4	7	1, 3, 5, 7, 10, 15, 20
	13	12	8	3, 7, 14, 28	0
	14	12	8	7	1, 3, 5, 7, 10, 15, 20
	15	12	12	3, 7, 14, 28	0
	16	12	12	7	1, 3, 5, 7, 10, 15, 20
	17	12	16	3, 7, 14, 28	0
	18	12	16	7	1, 3, 5, 7, 10, 15, 20

## Data Availability

This will be made available upon request through the corresponding author.

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
