# Peer review of "Experimental Study on PVA-MgO Composite Improvement of Sandy Soil"

_materials, 2022, doi:10.3390/ma15165609_

Round 1

Reviewer 1 Report

The research paper has been reviewed with the comments as follows;

1. It is reported that there was an inconsistent behavior of the compressive strength of the treated sand, what technical reasons have the authors for this behavior?

The English language of this paper is poor.

That is the optimized value of PVA needed to achieve the highest strength of the reconstituted sand?

What is the novelty of this research in relation to the civil engineering infrastructures and the sustainability of the materials?

It is natural for compressive strength to increase with curing age, what new findings did the authors make with the soil behavior?

The methodology of this research paper needs improvement to be replicated.

Extensive discussion of results related to the compressive strength behavior and the reactive state of the stabilization protocol is necessary to benefit future researchers.

Author Response

Comment No. 1: It is reported that there was an inconsistent behavior of the compressive strength of the treated sand, what technical reasons have the authors for this behavior?

Response: For pristine PVA stabilized samples, with the decreasing moisture content in the sample, PVA solution forms a complete elastic mesh, which enhances the strength of the sample.

For PVA and Mgo stabilized samples, activated magnesium oxide filled the pores between soil particles with cementitious materials in the process of wet-dry cycle. With the increasing number of wet-dry cycles, the active magnesium oxide fully reacts. It forms a compact and uniform consolidated substance among soil particles with PVA materials, which connects the sand particles into a whole.

Comment No. 2: The English language of this paper is poor.

Response: We are sorry about that, the problems for English language have been corrected.

Comment No. 3: That is the optimized value of PVA needed to achieve the highest strength of the reconstituted sand?

Response: 12%PVA is the best dosage, and the compressive strength is the highest under this dosage. When the content of PVA exceeds 12%, the sample can't keep its shape.

Comment No. 4: What is the novelty of this research in relation to the civil engineering infrastructures and the sustainability of the materials?

Response: In some cases, PVA can be used as a cementing material in civil engineering construction, which can provide sufficient strength. In terms of sustainability, PVA will not produce new reactions in sand or soil, and only provide the long chains to connect soil particles, which will not have a destructive impact on the environment.

Comment No. 5: It is natural for compressive strength to increase with curing age, what new findings did the authors make with the soil behavior?

Response: The loss of water can accelerate the generation of PVA gel, which can increase the strength of soil structure. If the water can be lost quickly by using reasonable curing methods, the required strength can be achieved in a short time. It doesn't need to be given enough temperature and humidity like cement, and it takes 28 days to reach it.

Comment No. 6: The methodology of this research paper needs improvement to be replicated.

Response: Thanks for the suggestion. In this study, strength is considered as the primary judgment factor of improvement effect and the focus is on unconfined compressive strength.

Comment No. 7: Extensive discussion of results related to the compressive strength behavior and the reactive state of the stabilization protocol is necessary to benefit future researchers.

Response: In the revised manuscript, the discussion section has been added.

Reviewer 2 Report

The article “Experimental study on PVA- MgO composite improvement of 2 sandy soil” is badly write and described. The English must be improved by a native English mother language, for example “was taken” ERROR, correct with “was collected”; “It can be seen…” is poor, change; “point” change with “area”; “picture” change with “image”.

For this reason, it very difficult to understand the manuscript, especially the Material and Methods section, which is very poor described.

In addition, no comparisons with other literature experiments are described. I think that authors must add a new chapter “DISCUSSION” to give comparison with they results with literature to give much more importance on their research work.

Only with major revision, the manuscript will be accepted.

Author Response

Comment No. 1: In the first paragraph, authors must specify the use of the sandy soil and its importance in which kinds of environment / work.

Response: The sandy soil slopes are widely distributed in southeast Tibet.

Comment No. 2: Add a short paragraph to introduce materials and methods before 2.1. chapter

Response: A short paragraph was added to describe the materials and methods.

Comment No. 3: Suggestions for amendments to 2.1.

Response: Thanks for the suggestion, the depth of soil sampling and the brand of PVA are described in detail. Subheading have been added to Figure 2.

Comment No. 4: "Weigh and record the weight of the sample" is it necessary to write it?

Response: As the reviewer suggested, the description of the test process has been modified.

Comment No. 5: Suggestions for amendments to 2.3.4.

Response: (1) “SEM” means scanning electron microscope.  “room temperature” means normal atmospheric temperature.(2)For dry-wet cycle test, the corresponding test flow was established based on the similar reference as follow:

“SONG Z Z, HAO S F, MEI H, et al. Strength characteristics of biopolymer modified sand under dry-wet cycle [J]. Acta Materiae Compositae Sinica, 2023, 40 (in Chinese). ”

Comment No. 6: For the suggestions in the results part.

Response: The explanation content is deleted and the discussion part is added. The picture is also been corrected.

Comment No. 6: For the suggestions in the conclusion part.

Response: “As can be seen from the above research” has been delete. The conclusion section is simplified.

Reviewer 3 Report

Overall the paper needs extensive revision to improve the presentation quality and the transfer of scientific knowledge. A few comments supporting this criticism are given. The authors need to correct other such needs, too.

The choice of PVA as a soil modifier needs more supporting arguments. In some industrial sectors there is a shortage of PVA material such that cost and availability might be concerns. Large volumes are needed for the type of application areas mentioned in the paper.

The conclusions are interesting and potentially important.

Comments:

At the end of the abstract, "but the already formed structure will be destroyed..." is difficult to interpret from the other information in the abstract. Furthermore, this statement should be made as a separate sentence, not as a run on of the previous sentence.

Some revisions are needed to improve readability. Take the first sentence, for example, it should be "coarse-grained soil slopes have...".  What is meant by "other environments"? It is fair to say "improved", instead of "optimized", since improvements are what is needed. Due to the type of problem considered, "optimization" sounds far-reaching.

As it is a current topic of interest, bio-remediation should be mentioned in the first paragraph as a potential technique for soil improvement. Relevant papers could be cited.

At the start of the second paragraph, it is not advisable to localize the discussion to any specific geographical region. Southeast Tibet could be mentioned as an example, but the current paper places focus on this region only.

The motivations for using PVA should be described in the introduction.

Furthermore the introduction should put the authors' research into focus relative to current other attempts to modify soil behavior. The citations given by the authors provide a general picture, but do not properly motivate the research or explain its novelty compared to other recent work.

The vertical axis of Figure 1 should be "percentage passing"

The properties of the PVA are introduced in the section on "sandy soil", which does not make sense. Similarly the caption for Figure 2 is inappropriate.

The section heading "improved materials" is not representative of the section contents. "soil additive materials" or something like that would fit better.

The particle sizes of the magnesium oxide are said to be "evenly distributed". Is this on an arithmetic scale, a logarithmic scale, or some other scale?

The following text (in quotes) does not make sense. The paper presents many opportunities for clarification and revision. "When PVA dosage was determined to be 6%, 8%, 10% and 12%, and PVA dosage was determined to be 12%, the dosage of active magnesium oxide was 4%, 8%, 12% and 16% respectively, and the curing age was."

How was the PVA solution added to the soil samples?

Were the samples consolidated in any way prior to compressive strength testing?

In Figure 3b should the third legend entry be 10%?

The following (in quotes) needs explanation. "It can be seen from the figure that the 154 unconfined compressive strength of improved soil increases obviously with the increase of PVA content, because the sample contains more water at this time, and the lubrication of water among soil particles is the main control factor. "

The section 3.5 on microscopic analysis presents SEM images for each of the cases considered. This part should be made more concise. It might be helpful to present only a subset of these figures.

In conclusion 4 the permeability coefficient values need units.

The above is just a partial listing of needs for revision. It is the authors’ responsibility to attend to those not covered in this review, as well.

Author Response

Response to the comments of Reviewer #3

Comment No. 1: At the end of the abstract, "but the already formed structure will be destroyed..." is difficult to interpret from the other information in the abstract. Furthermore, this statement should be made as a separate sentence, not as a run on of the previous sentence.

Response: Thanks for the suggestion, the abstract has been revised.

Comment No. 2: Some revisions are needed to improve readability. Take the first sentence, for example, it should be "coarse-grained soil slopes have...". What is meant by "other environments"? It is fair to say "improved", instead of "optimized", since improvements are what is needed. Due to the type of problem considered, "optimization" sounds far-reaching.

Response: Thanks for the suggestion, the existing problems in expression have been modified.

Comment No. 3: As it is a current topic of interest, bio-remediation should be mentioned in the first paragraph as a potential technique for soil improvement. Relevant papers could be cited.

Response: Thanks for the suggestion, the literature citation of bioremediation has been added.

Comment No. 4: At the start of the second paragraph, it is not advisable to localize the discussion to any specific geographical region. Southeast Tibet could be mentioned as an example, but the current paper places focus on this region only.

Response: Thanks for the suggestion. Because it is temporarily impossible to obtain sand samples from other areas, southeast Tibet has been taken as an example.

Comment No. 5: The motivations for using PVA should be described in the introduction. Furthermore the introduction should put the authors' research into focus relative to current other attempts to modify soil behavior. The citations given by the authors provide a general picture, but do not properly motivate the research or explain its novelty compared to other recent work.

Response: Thanks for the suggestion, the introduction has been revised.

Comment No. 6: The vertical axis of Figure 1 should be "percentage passing"

Response: Thanks for the suggestion, figure 1 has been modified.

Comment No. 7: The properties of the PVA are introduced in the section on "sandy soil", which does not make sense. Similarly the caption for Figure 2 is inappropriate.

Response: Thanks for the suggestion, the corrections have been made.

Comment No. 8: The section heading "improved materials" is not representative of the section contents. "soil additive materials" or something like that would fit better.

Response: Thanks for the suggestion. The section heading "improved materials" is deleted, and all materials used in the research are classified as "materials"

Comment No. 9: The particle sizes of the magnesium oxide are said to be "evenly distributed". Is this on an arithmetic scale, a logarithmic scale, or some other scale?

Response: In the revised manuscript, the content is modified to "the particle diameter distribution is in the range of 0.14-40.15 μm".

Comment No. 10: The following text (in quotes) does not make sense. The paper presents many opportunities for clarification and revision. "When PVA dosage was determined to be 6%, 8%, 10% and 12%, and PVA dosage was determined to be 12%, the dosage of active magnesium oxide was 4%, 8%, 12% and 16% respectively, and the curing age was."

Response: Thanks for the suggestion, table 2 has been added to clearly show the test scheme.

Comment No. 11: How was the PVA solution added to the soil samples? 

Response: The PVA solution is weighed and added into the soil according to the required proportion. Proportion refers to the ratio of PVA solution to soil weight.

Comment No. 12: Were the samples consolidated in any way prior to compressive strength testing?

Response: This test is unconfined compression test without consolidation, and all samples are compressed according to the density designed in the test.

Comment No. 13: In Figure 3b should the third legend entry be 10%?

Response: Thanks for the suggestion, the content has been corrected.

Comment No. 14: The following (in quotes) needs explanation. "It can be seen from the figure that the 154 unconfined compressive strength of improved soil increases obviously with the increase of PVA content, because the sample contains more water at this time, and the lubrication of water among soil particles is the main control factor. "

Response: The discussion part was added to show that there is more moisture in the sample.

Comment No. 15: The section 3.5 on microscopic analysis presents SEM images for each of the cases considered. This part should be made more concise. It might be helpful to present only a subset of these figures.

Response: We think that keeping all the pictures may show the internal structure of the improved sand in detail.

Comment No. 16: In conclusion 4 the permeability coefficient values need units.

Response: Thanks for the suggestion, we simplified the conclusion.

Round 2

Reviewer 1 Report

Comments have been responded to substantially 

Author Response

 Thanks again to the hard work of the reviewers!

Reviewer 2 Report

Accept for publication

Author Response

(The authors gave the same response as above.)

Reviewer 3 Report

Presentation quality has improved, but the authors still need to go through the paper to correct grammatical and syntactic errors. The first sentence in the abstract is a prime example of the needs for such revision.

About the first sentence in the abstract, the previous round of comments included the suggestion not to localize the research discussion to Tibet at the start of the abstract. This should be common sense for submissions to this type of international journal. The authors results have general applicability to these types of soils, which has to be a main point. It is relatively simple to open the abstract with reference to general behavior and then localize it to Tibet later in the paper.

It was also suggested to briefly discuss and cite relevant literature on bioremediation or biomodification of sandy soils in the introduction. The authors' response letter says this was done, but the corresponding changes are not evident in the paper. A couple references were added in the introduction, but those do not pertain to the hot topic of bioremediation as it is currently understood.

In newly added Table 2, Mgo should be MgO.

Author Response

Response to the comments of Reviewer #3

Comment No. 1: Presentation quality has improved, but the authors still need to go through the paper to correct grammatical and syntactic errors. The first sentence in the abstract is a prime example of the needs for such revision.

Response: Thanks for the suggestion. The grammatical and syntactic errors are corrected.

Comment No. 2: About the first sentence in the abstract, the previous round of comments included the suggestion not to localize the research discussion to Tibet at the start of the abstract. This should be common sense for submissions to this type of international journal. The authors results have general applicability to these types of soils, which has to be a main point. It is relatively simple to open the abstract with reference to general behavior and then localize it to Tibet later in the paper.

Response: Thanks for the suggestion. The abstract is modified.

Comment No. 3: It was also suggested to briefly discuss and cite relevant literature on bioremediation or biomodification of sandy soils in the introduction. The authors' response letter says this was done, but the corresponding changes are not evident in the paper. A couple references were added in the introduction, but those do not pertain to the hot topic of bioremediation as it is currently understood.

Response: Thanks for the suggestion. We only added a few quotations to the introduction to indicate that it was related to bioremediation in the last version. Now, we have added a discussion on bioremediation to the introduction.

Comment No. 4: In newly added Table 2, Mgo should be MgO.

Response: Thanks for the suggestion, we made a correction.